# OpenReview forum: "On-the-fly Preference Alignment via Principle-Guided Decoding"
_ICLR.cc/2025/Conference — ICLR 2025 Poster_

### Official Review · Reviewer_dQvo · 2024-11-01

**Soundness:** 2
**Presentation:** 2
**Contribution:** 3
**Rating:** 6
**Confidence:** 3

**Summary:**

This work introduces a decoding method called OPAD that aligns model outputs with human preferences during inference. OPAD modifies model predictions to ensure compliance with human preferences. Compared to algorithms like DPO/PPO, OPAD achieves better performance without requiring additional training computational costs, which is very exciting.

**Strengths:**

The authors introduce an alternative objective that maximizes the KL divergence between constrained and unconstrained policies during decoding. This approach quantifies the model's adherence to target preferences, thereby determining the reward in the alignment process. This method adjusts token prediction probabilities to promote preference compliance.

The major advantage of this paper is that OPAD outperforms DPO/PPO algorithms and Best-of-N methods with significantly fewer computational resources and requires no additional training.

**Weaknesses:**

Recent research has focused on alignment during the decoding phase, such as [1][2][3][4], and the authors should discuss these works.

Some content lacks clarity, such as what constitutes a Principle in Alignment?  Additionally, to my knowledge, for models like Llama-3.1, if Principles are set in the system prompt, they can follow the described text. Therefore, experimenting with newer SOTA open-source models would help strengthen the paper's robustness.

The paper lacks theoretical explanation for why the proposed method is superior to placing Principles in ICL (In-Context Learning).

---

[1] DeAL: Decoding-time Alignment for Large Language Models

[2] Decoding-Time Language Model Alignment with Multiple Objectives

[3] Reward Steering with Evolutionary Heuristics for Decoding-time Alignment

[4]  Personality Alignment of Large Language Models

**Questions:**

How are Principles set in Summarization and HH-RLHF?

---

> ### Author Response · Authors · 2024-11-17
> **Author Response to Reviewer dQvo (Part 1/2)**
>
> We appreciate the reviewer’s constructive comments and detailed review. We also thank the reviewer for pointing out some limitations. We hope that the responses can address the reviewer’s concerns and we welcome further discussion.
>
> >"Recent research has focused on alignment during the decoding phase, such as [1][2][3][4], and the authors should discuss these works."
>
> Thanks for the reviewer’s suggestion to include a more detailed discussion of the related works. We have modified the Related Work section accordingly in Section 2.2 (line 149).
>
> >"Some content lacks clarity, such as what constitutes a Principle in Alignment?"
>
> Thank you for pointing out the ambiguity in this terminology. In our context, a principle refers to a clear and descriptive guideline that encapsulates the underlying preference or desired behavior. This terminology is similarly employed in [1]. The purpose of principles is to explicitly instruct the model on what types of responses or practices are preferred and aligned with specific goals or values.
>
> To improve clarity, we have revised Section 4.1 to include a Principle Curation part:
>
> *Throughout the experiments, we curate principles based on task-specific heuristics. For general alignment tasks that an instruction-tuned model typically handles well (such as helpfulness and harmlessness), we carefully curate the principle $c$ to better illustrate what this universal preference means, thus granting the model a better understanding of these universal principles. For personal preference alignment tasks, the principle $c$ is a direct statement to ask the model to behave in a desired manner (e.g. Please act as if you are an experienced researcher).*
>
> >"if Principles are set in the system prompt, they can follow the described text…"
>
> Thank you for the insightful suggestion. Adding principles via the system prompt is indeed a form of prompt engineering, where the system role provides global context for how the model should respond. In our experiments with Mistral and Vicuna (our base models), principles were set via system prompts, followed by user/assistant conversations. This approach corresponds to the Principle Prompting (PP) baseline. The results show that while system prompts can guide the model, they often fail to fully enforce the desired principles, where OPAD demonstrates a clear advantage.
>
> Following your suggestion, we extended our experiments to the newer SOTA open-source model, **Llama-3.1-8B-Instruct**, where principles were also added via system prompts. Our findings are as follows:
>
> **On HH_RLHF**: Llama-3.1-8B-Instruct, fine-tuned with SFT and RLHF for helpfulness and safety, establishes PP as a strong baseline. However, OPAD demonstrates an edge by further enhancing alignment with the desired principles.
>
> **On DSP**: OPAD shows a notably greater advantage, excelling in aligning model outputs with target principles for personalized tasks, even when applied to this highly capable base model.
>
> These results further highlight OPAD’s ability to improve adherence to principles, particularly for specialized or personalized tasks, even when using state-of-the-art models.
> | OPAD vs. | HH-RLHF (win) | HH-RLHF (lose) | DSP (win) | DSP (lose) |
> |----------|---------------|----------------|-----------|------------|
> | DP       | 16.5          | 13.0           | 72.5      | 12.0       |
> | **PP**       | **19.5**         | **16.5**          | **35.0**      | **15.0**      |
> | ICL      | 30.5          | 16.5           | 42.5      | 15.3       |
> |Bo$N$ | 19.5         | 21.8          | -      | -     |
> | self-CD       | 27.5          | 22.5           | 31.0      | 21.0       |
> | LA       | 18.0          | 14.5           | 20.0      | 17.5       |
>
> (Please see part 2 of our response.)

---

> > ### Author Response · Authors · 2024-11-20
> > **Author Response to Reviewer dQvo (Part 2/2)**
> >
> > (Please read part 1 of our response first.)
> >
> > > "The paper lacks theoretical explanation for why the proposed method is superior to placing Principles in ICL"
> > >
> >
> > ICL techniques are known to fall short in handling specification-heavy tasks [2] and predicting complex or nuanced contexts [3].
> > The goal of OPAD is to go beyond the limitations of ICL by "reinforcing" the underlying principle to the model. This allows the model to better adhere to the preference principles without requiring retraining or fine-tuning, particularly when the model itself lacks the capability to perfectly follow instructions. By directly incorporating the principles into the decoding process, OPAD provides additional guidance during inference, addressing the weaknesses of ICL in dealing with tasks that require deeper understanding or stricter adherence to instructions.
> >
> > Our experiments with models across different scales and capabilities indicate that OPAD can generalize well and is very effective on a wide range of mid-scale models (e.g., Vicuna-7B and Vicuna-13B), where it effectively “reinforces” principles to improve alignment (thus superior to ICL techniques). Please find more analysis for this in Appendix C.
> >
> > [1]. Gao, Songyang, et al. "Linear Alignment: A Closed-form Solution for Aligning Human Preferences without Tuning and Feedback." *arXiv preprint arXiv:2401.11458* (2024).
> >
> > [2]. Peng, Hao, et al. "When does In-context Learning Fall Short and Why? A Study on Specification-Heavy Tasks." *arXiv preprint arXiv:2311.08993* (2023).
> >
> > [3].Kossen, Jannik, Yarin Gal, and Tom Rainforth. "In-context learning learns label relationships but is not conventional learning." *The Twelfth International Conference on Learning Representations*. 2024.
> >
> > > "How are Principles set in Summarization and HH-RLHF?"
> > >
> >
> > The detailed principles are demonstrated in Appendix E and we display it here for your convenience:
> >
> > For Summarization:
> >
> > *Make sure the summary is concise and comprehensive. The summary should capture the main points and key details of the text while conveying the OP’s intended meaning accurately. The length of the summary should be appropriate to capture the main points and key details of the text, without including unnecessary information or becoming overly long.*
> >
> > For HH-RLHF:
> >
> > *Please adhere to the following principles. Avoid factual inaccuracies as much as possible.
> > Refrain from providing answers if the user’s request poses potential security concerns, and
> > provide relevant explanations and guidance instead. If the previous context did not address
> > the user’s issue, continue attempting to answer and resolve it. Stay on track with the original
> > discussion and avoid introducing unnecessary off-topic information. Enhance answers by incorporating additional background information to assist users in understanding and grasping
> > the content.*
> >
> > Again, we thank the reviewer for the detailed review. We hope that our reply has addressed any remaining concerns. Please let us know if there are any further changes you would like to see or if there is anything else that we can clarify.

---

> > > ### Comment · Reviewer_dQvo · 2024-11-20
> > >
> > > Thank you for your comprehensive response! I have carefully reviewed your revised manuscript and responses, and I'm pleased to see that most of my concerns have been thoroughly addressed.
> > >
> > > However, I noticed in Table 1 that BoN and Self-Contrastive Decoding achieve the best and second-best performance respectively. To strengthen the robustness of your findings, I would recommend conducting additional experiments with Llama-3 models for these methods. This would not only provide valuable insights into how your approach compares across different model architectures but also enhance the paper's contribution to the field.
> > >
> > > Additionally, while your responses contain compelling experimental results, I strongly encourage you to incorporate these findings directly into the manuscript rather than keeping them solely in the response letter. This integration would significantly enhance the paper's comprehensiveness and provide readers with a more complete understanding of your work.
> > >
> > > I look forward to seeing these additions in your final version!

---

> > > > ### Author Response · Authors · 2024-11-27
> > > > **All additions ready in the finial revision**
> > > >
> > > > Dear reviewer dQvo,
> > > >
> > > > We hope this message finds you well.
> > > >
> > > > We sincerely appreciate the time and effort you have dedicated to reviewing our paper. Your valuable insights and constructive feedback have been instrumental in improving our manuscript.
> > > >
> > > > As per your suggestion, we have reorganized the manuscript and incorporated an in-depth analysis of the Llama3 family in Section 4.4 (highlighted in blue, Line 438). This addition provides a comprehensive comparison across different model architectures, which we believe adds significant value and strengthens the overall narrative of our work.
> > > >
> > > > With the revision deadline approaching, we would be immensely grateful if you could review our updates and engage with our rebuttal during the remaining discussion period. We are committed to addressing any further concerns or suggestions you may have.
> > > >
> > > > Many thanks in advance
> > > >
> > > > Best regards,

---

> > > > > ### Comment · Reviewer_dQvo · 2024-11-29
> > > > > **Thanks for your response!**
> > > > >
> > > > > Thank you for updating the paper! I believe our discussion has helped strengthen its quality. I would be delighted to see this paper accepted!

---

> > > > > > ### Author Response · Authors · 2024-11-29
> > > > > > **Thanks for the reply!**
> > > > > >
> > > > > > Thanks for your reply. We really appreciate your valuable feedback to help improve the quality of our manuscript, and that you would be delighted to see this paper accepted!

---

> ### Author Response · Authors · 2024-11-21
> **Author Response to Reviewer dQvo**
>
> Thank you for your quick response and valuable suggestions for improving our manuscript!
>
> First, we would like to apologize for the typo in our earlier response, where the "CD" baseline is actually the Self-Contrastive Decoding (self-CD) method (this has now been corrected, as well as Bo$N$ for HH-RLHF). For highly personalized tasks such as DSP, where no specific reward models are directly available, we omitted the Bo$N$ baseline for personalized alignment tasks, following the methodology outlined in the main paper. Since the base model is inherently aligned with helpfulness and safety topics and we use a very strong reward model that was trained on the HH-RLHF task to select the best answer, Bo$N$ demonstrates strong performance. However, Bo$N$ is resource-intensive, as it requires sampling multiple responses for ranking, making it less efficient in practical applications. In contrast, OPAD achieves reasonable results while being significantly more resource-efficient.
>
> In response to your suggestion, we have conducted additional and comprehensive experiments with Llama-3.2-1B-Instruct, Llama-3.2-3B-Instruct, and Llama-3.1-8B-Instruct on both HH-RLHF and DSP tasks. These findings have been directly incorporated into the manuscript to enhance its robustness and contribution to the field. A detailed analysis is provided in Appendix C (line 804, highlighted in blue for your convenience). Please note that while the results are currently consolidated in the appendix, we plan to distribute them throughout the manuscript in the final version to better integrate them with the main text.
>
> We greatly appreciate your feedback and please let us know if there are any further changes you would like us to make.

---

### Official Review · Reviewer_Se9h · 2024-11-03

**Soundness:** 3
**Presentation:** 2
**Contribution:** 3
**Rating:** 6
**Confidence:** 4

**Summary:**

This paper introduces On-the-fly Preference Alignment via Principle-Guided Decoding (OPAD), addressing the challenge of aligning language models with human preferences without requiring expensive fine-tuning or extensive data collection. The method works by designing a principle-guided reward function based on a surrogate optimization solution and directly modifying model predictions during inference to ensure adherence to target principles. Experimental results demonstrate that OPAD achieves competitive or superior performance compared to existing baselines across both general and personalized alignment tasks.

**Strengths:**

- The paper introduces a novel tuning-free alignment approach that uniquely leverages KL divergence between constrained and unconstrained policies during inference.
- The method demonstrates robust empirical validation through comprehensive experiments across multiple datasets and strong performance against RLHF, DPO, and other baseline methods.

**Weaknesses:**

- The derivation of the surrogate optimization solution relies on an overly strong condition, namely 'The constraint c aligns well with the data distribution $P_{\text{data}}$'. In lines 214-215, there is a statement saying 'Direct optimization is infeasible since we have no access to $P_{\text{data}}$'. However, the authors did not explain how to design principles without access to $P_{\text{data}}$, and how to verify and guarantee this condition holds.
In fact, even if the constraint $c$ partially aligns with $P_{\text{data}}$, it could still lead to bad alignment. For example, let $P_{\text{data}}$ be 'professional medical advice', $P$ generate health-related statements with broad coverage but low accuracy, and $c$ be 'use formal medical terminology'. Maximizing $\text{KL}(P_c||P)$ might lead to technically-worded but meaningless or incorrect medical statements.
Therefore, the alignment quality of the principle is crucial. However, designing well-aligned principles without access to $P_{\text{data}}$ seems almost impossible; at least a surrogate $P_{\text{data}}$ is needed for guidance and principle evaluation. I believe the authors' experiments actually had access to $P_{\text{data}}$ (i.e., we clearly know we're dealing with the harmless and helpful aspects of hh-rlhf), so the designed principles could be well-aligned. But this contradicts the assumption of having no access to $P_{\text{data}}$.

- The proposed method lacks computational efficiency, running twice as slow as standard model decoding. While the tuning-free approach eliminates training costs, these costs are typically one-time investments, whereas inference requires continuous computational overhead. This computational burden limits the method's practical applications. Additionally, I recommend including computational efficiency comparisons with other in-context alignment methods to demonstrate whether OPAD offers advantages over similar approaches

-  For a paper focusing on decoding-time alignment, the Methodology section lacks both a complete description of the decoding steps and clear connections to the method overview presented in Figure 2, which reduces the paper's clarity. I recommend relocating the 'Relation with the residual EBMs' section to the later analysis and discussion portion, as it isn't a direct methodology description, and enhancing the method section with either a complete description of the decoding process or pseudocode to improve clarity.

**Questions:**

- Why do more steps hurt performance? Theoretically, global rewards are necessary because the text that aligns locally can violate principles globally. For example, for a medical advice principle, 'Take aspirin for headaches. Then take ibuprofen for fever.' While each 2-token window appears medically sound, this advice globally violates medical safety.

---

> ### Author Response · Authors · 2024-11-17
> **Author Response to Reviewer Se9h (Part 1/2)**
>
> We want to thank the reviewer for the insightful feedback and the detailed comments that helps us better develop and validate the proposed approach. We hope that the responses can address the reviewer’s concerns and we truly welcome further discussion.
>
> > "designing well-aligned principles without access to $P_{data}$ seems almost impossible; at least a surrogate $P_{data}$ is needed for guidance and principle evaluation."
>
> We apologize for any confusion caused by our earlier statement regarding “we have no access to $P_{data}$”. To clarify, we meant that we do not have direct access to the **training data** itself. We agree with the reviewer that designing well-aligned principles requires prior knowledge of the task, goal, or preferences involved. And we have revised the wording in Proposition 1 (Line 219 in blue) to ensure clarity and precision.
>
> >"how to design principles without access to $P_{data}$, and how to verify and guarantee this condition holds."
>
> As clarified above, we need prior knowledge of the task, goal, or preferences when designing the principles, which is usually a clear and descriptive guideline that encapsulates the underlying preference or desired behavior. In our experiments, the preference principles are constructed using heuristics, and we have included a detailed explanation of this process titled Principle Curation in Section 3.3 (Line 316) as well as here for your convenience:
>
> *In our experiments, we curate principles using task-specific heuristics. For general alignment tasks (e.g., helpfulness and harmlessness), we define the principle $c$ to clearly communicate universal preferences, helping the model understand these concepts. For personal preference alignment tasks, the principle $c$ directly instructs the model to act in a desired way (e.g., "Please behave as if you are an experienced researcher"). The specific principles for each task are provided in Appendix E as well.*
>
> >"In fact, even if the constraint $c$ partially aligns with $P_{data}$, it could still lead to bad alignment."
>
> Typically, inference-time algorithms (without relying on RAG or additional sources) assume that the model already retains the necessary knowledge to answer the query, and our goal is to elicit it more effectively. The base model generating “health-related statements but with low accuracy” means the knowledge base preserved in the model is not sufficient, and re-training or fine-tuning may be required in this case.
>
> As stated in Appendix C, we want to emphasize that the **best use case for our method** lies in scenarios where the model retains sufficient domain knowledge, but fails to follow the instructions perfectly. For such models, our approach effectively “reinforces” the principles to guide the model to respond in an aligned and preferred manner without requiring fine-tuning. Conversely, for smaller models that lack the foundational knowledge, or for larger models that already follow instructions well, the benefit of inference-time algorithms diminishes.
>
> >"the Methodology section lacks both a complete description of the decoding steps…"
>
> This is a very pertinent point and we appreciate the reviewer for the suggestion to improve the clarity of this manuscript. We have relocated the 'Relation with the residual EBMs' section to the later analysis and provided a detailed decoding algorithm instead to improve clarity. (line 268 in blue)
>
> (Please see part 2 of our response.)

---

> > ### Author Response · Authors · 2024-11-17
> > **Author Response to Reviewer Se9h (Part 2/2)**
> >
> > (Please read part 1 of our response first.)
> >
> > >"The proposed method lacks computational efficiency…."
> >
> > We actually have included the computation efficiency statistics in Table 5 in our initial submission. As the reviewer noted, OPAD incurs approximately twice the time (due to two forward passes) compared to standard inference, while maintaining similar memory usage during decoding. We can also merge the two forward passes into a single batch for parallel processing to speed it up. A similar computational demand is seen with self-CD and LA, which also require two forward passes during decoding. BoN, on the other hand, involves sampling multiple times, which increases resource consumption.
> >
> > Next, we analyze the scenarios in which leveraging OPAD (which incurs additional computational overhead) has an efficiency advantage over DPO training (a one-time computational investment). Assume that a forward pass for the base model requires B FLOPs, while a backward pass requires 2B FLOPs. Let the training set size be N and the inference set size be M.
> > For DPO, which involves forward passes for both positive and negative samples, the total FLOPs can be estimated as approximately (N+N+2N+M)B. For OPAD, the total computational cost is approximately 2MB FLOPs. From this, we can conclude that OPAD is more efficient when (4N+M)>2M, or equivalently when M<4N. Given that N is typically large (often in the tens of thousands), OPAD is generally a more computationally efficient choice.
> >
> > Besides, the training process might involve trial-and-error and various optimization techniques, which may require multiple experiments to achieve the desired results. However, inference-time algorithms can be used directly, with relatively lower risks and resource requirements. In practice, we could consider a scenario where training is only conducted when inference fails to meet the necessary requirements.
> >
> > >"Why do more steps hurt performance? Theoretically, global rewards are necessary because the text that aligns locally can violate principles globally."
> >
> > Thank you for your thoughtful feedback. We appreciate your point regarding the importance of global rewards to ensure that locally aligned text does not violate principles on a global scale.
> >
> > In our initial experiments, we observed that simply extending the reward computation over more time steps led to a decrease in performance. This may be because, in sequential generation tasks, the influence of earlier tokens on the current step often diminishes over time, and appropriate weighting is required to accurately reflect the varying levels of importance in achieving global alignment. To verify this, we introduced a discount factor, specifically choosing a discount value of 0.6, and aggregated the reward over 4 time steps. The results showed that aggregating over more time steps did not provide additional alignment benefits while incurring higher computational costs compared to a simple two-step design.
> >
> >
> > | Time steps | OPAD Win vs. PP | OPAD Lose vs. PP |
> > |------------|------------------|------------------|
> > | 2          | 35.8            | 23.5            |
> > | 3          | 32.9            | 23.0            |
> > | 4          | 33.1            | 22.8            |
> >
> > The reason may be that the nature of language modelling means that prior information is already incorporated when decoding the current token, strictly enforcing a global adjustment can result in a coarse-grained alignment that is not as precise as a 2-token adjustment. Using two tokens is sufficient for effective modeling and demonstrates transferable characteristics across different tasks with less computational cost. Based on this consideration, we chose the two-token design.
> >
> > We hope these changes address the remaining concerns and are sufficient for you to be able to back acceptance. If there are any other confusion you would like us to clarify or other improvements you would like us to implement, please let us know – we would be glad to do so.

---

> > > ### Comment · Reviewer_Se9h · 2024-11-19
> > >
> > > Thank you for your thorough responses. My concerns have been well addressed, so I've adjusted the score accordingly.

---

> > > > ### Author Response · Authors · 2024-11-19
> > > > **Author Response**
> > > >
> > > > Thanks for your response and we really appreciate that! We are glad your concerns have been well addressed, and the score has been adjusted accordingly!

---

### Official Review · Reviewer_mD8g · 2024-11-04

**Soundness:** 2
**Presentation:** 3
**Contribution:** 3
**Rating:** 6
**Confidence:** 4

**Summary:**

This paper introduces On-the-fly Preference Alignment via Principle-Guided Decoding (OPAD), a novel approach for aligning large language model outputs with human preferences without requiring resource-intensive fine-tuning or retraining. Unlike RLHF, which rely on substantial data and computational resources, OPAD achieves preference alignment directly during inference. The method leverages a principle-guided reward function derived from a surrogate solution to an otherwise intractable optimization problem, allowing it to steer model predictions towards desired behaviors dynamically. OPAD demonstrates efficiency and effectiveness across general and personalized alignment tasks, often outperforming established baselines.

**Strengths:**

- OPAD offers a tuning-free alignment method inspired by the objectives of reinforcement learning, allowing it to align model outputs with human preferences without the need for additional tuning.
- The framework  can be applied to various tasks with minimal adjustments, making it a flexible solution for different alignment needs.
- Despite not requiring any model training, OPAD demonstrates competitive performance, often rivaling or surpassing traditional, training-intensive alignment methods.

**Weaknesses:**

- In some cases, there is a high percentage of ties instead of clear wins or losses. How should we interpret this outcome?
- Since your method does not involve any model training, it would be valuable to demonstrate its performance on larger, more capable models to assess the trend and generalizability of your approach across different model scales.
- How does your method perform when LLMs do not achieve sufficient performance, such as smaller LLMs?

**Questions:**

- Would it be more appropriate to rename Section 5 from 'Discussion' to 'Conclusion' for clarity?

---

> ### Author Response · Authors · 2024-11-17
> **Author Response to Reviewer mD8g (Part 1/2)**
>
> We thank the reviewer for the detailed review and constructive comments. Below are our clarifications to all the questions. We hope that the responses can address the reviewer’s concerns and we truly welcome further discussion.
>
> >"In some cases, there is a high percentage of ties instead of clear wins or losses"
>
> The high percentage of ties observed in some cases is primarily due to the inherent **positional bias** in using LLMs like GPT-4 as evaluators. Studies, such as [1], show that GPT-4 tends to favor the first displayed response, even when the order of candidates is changed. To mitigate this bias, we use a pairwise comparison approach, which is standard in preference evaluation pipelines.
>
> Despite this mitigation, GPT-4 often provides a "tie" judgment or shows inconsistent preferences when the order of responses is switched, especially when the quality gap between the responses is small. This results in higher tie rates.
>
> In instances where the tie rate is high, it indicates that the baseline method is closely comparable to our proposed OPAD method, leading to GPT-4's inconsistent preferences upon order reversal (e.g., BoN in Table 1). However, it's important to note that in most cases, the Win rates for our method still dominate.
>
> Besides, high tie rates are not uncommon in preference evaluation literature, as observed in [2] (40-50% in Figure 2), [3] (around 60% in Figure 5), and [4] (50-100% in Figures 4 & 5).
>
> [1]. Wang, Peiyi, et al. "Large language models are not fair evaluators." *arXiv preprint arXiv:2305.17926* (2023).
>
> [2].Gao, Songyang, et al. "Linear Alignment: A Closed-form Solution for Aligning Human Preferences without Tuning and Feedback." *arXiv preprint arXiv:2401.11458* (2024).
>
> [3]. Zeng, Yongcheng, et al. "Token-level Direct Preference Optimization." *arXiv preprint arXiv:2404.11999* (2024).
>
> [4]. Shi, Chenyu, et al. "Navigating the overkill in large language models." *arXiv preprint arXiv:2401.17633* (2024).
>
> >"it would be valuable to demonstrate its performance on larger, more capable models to assess the trend and generalizability of your approach across different model scales"
>
> Thanks for the valuable suggestion.  We conducted additional experiments on models of varying capabilities, including Vicuna-13B, Vicuna-33B (more capable), and Pythia-2.8B (less capable), across a general alignment task (HH-RLHF) and a personalized task (DSP).
>
> | HH-RLHF  | Pythia-2.8b|Vicuna-7b     | Vicuna-13b    | Vicuna-33b    |
> |-------------------------|----------------|---------------|---------------|---------------|
> | OPAD (ours) vs.        | win / lose       | win / lose      | win / lose      | win / lose      |
> | DP                      | 31.5 / 29.3    | 51.3 / 23.8   | 51.8 / 23.0   | 39.3 / 23.3   |
> | PP                      | 32.5 / 24.8    | 43.5 / 27.5   | 46.8 / 25.3   | 35.5 / 27.3  |
> | ICL                     | 28.8 / 27.3    | 53.3 / 24.0   | 49.5 / 25.5   | 48.8 / 23.8   |
> | self-CD                 | 23.5 / 31.0    | 39.0 / 35.5   | 50.8 / 27.8   | 46.2 / 30.0   |
> | LA                      | 30.8 / 22.5    | 36.0 / 33.25  | 33.5 / 36.0   | 31.0 / 30.3   |
>
> | DSP  | Pythia-2.8b    | Vicuna-7b     | Vicuna-13b    | Vicuna-33b    |
> |--------------------------|----------------|---------------|---------------|---------------|
> | OPAD (ours) vs.         | win / lose       | win / lose      | win / lose      | win / lose      |
> | DP                       | 26.3 / 30.8    | 74.8 / 7.0    | 71.5 / 9.5    | 63.5 / 10.0   |
> | PP                      | 30.3 / 22.8    | 56.0 /  7.5   |  46.5 / 15.3  | 39.5 / 12.5   |
> | ICL                      | 36.0 / 26.0    | 74.0 / 8.5    | 66.8 / 15.3   | 66.3 / 13.3   |
> | self-CD                  | 35.3 / 34.3    | 39.5 / 35.6   | 45.3 / 29.8   | 30.5 / 29.5   |
> | LA                       | 38.0 / 20.3    | 35.5 / 19.5   | 30.75 / 24.3  | 25.3 / 21.8   |
>
> The results with PP (principle prompting) clearly illustrate key trends in the scalability and generalizability of OPAD. We observe an increasing then decreasing trend in OPAD's relative effectiveness as the model scale increases:
>
> (Please see part 2 of our response.)

---

> ### Author Response · Authors · 2024-11-17
> **Author Response to Reviewer mD8g (Part 2/2)**
>
> (Please read part 1 of our response first.)
>
> 1. **Performance on mid-scale (less capable) models:** OPAD is most effective on mid-scale models (e.g., Vicuna-7B and Vicuna-13B) that retain sufficient domain knowledge, but fails to follow the instructions perfectly. OPAD manages to effectively “reinforce” principles to improve alignment. In practical scenarios, 7B and 13B models are widely used due to their balance between capability and resource efficiency. OPAD can help these models adapt better to task-specific requirements during decoding without requiring fine-tuning.
> 2. **Performance on larger (more capable) models:** As model scales up, direct prompting with principles (PP) becomes a stronger baseline.  Larger models are inherently better at following instructions, exhibiting diminished marginal benefits from OPAD.
> 3. **Performance on smaller (weak) models:** For weak models (Pythia-2.8B), they often lack the foundational ability or knowledge to respond appropriately to user queries, making the “reinforcement” provided by OPAD less impactful. In these cases, fine-tuning is ultimately required.
>
> We have incorporated the findings and analysis directly into the manuscript in Section 4.4, Line 426 (highlighted in blue) to showcase the generalization capability of the proposed method. This addition offers clear guidance on the optimal use cases for our approach. Thank you again for your valuable feedback!
>
>
> >"How does your method perform when LLMs do not achieve sufficient performance, such as smaller LLMs?"
>
> To explore this, we evaluated our method using the SFT-ed Pythia-2.8B model, which is a smaller and much weaker LLM. The result, as indicated above, shows that OPAD is less effective when applied to models that do not inherently achieve sufficient performance, since the reinforcement signal brought by our method cannot fully compensate for the model’s inherent weaknesses. However, compared to other inference-time baselines, OPAD still demonstrates an advantage.
>
> >"Would it be more appropriate to rename Section 5 from 'Discussion' to 'Conclusion' for clarity?"
>
> Yes, we have renamed Section 5 to “Conclusion” for clarity. Thanks again for the valuable feedback!
>
> Thanks again very much for the review. We hope that our reply has addressed your concerns. Please let us know if there are any further changes you would like to see or if there is anything else that we can clarify.

---

> ### Author Response · Authors · 2024-11-24
> **Discussion Period Ends Soon**
>
> Dear Reviewer mD8g,
>
> We hope this message finds you well.
>
> We appreciate your time and effort devoted to reviewing our paper and your insights and feedback have been truly invaluable to improving our manuscript.
>
> The discussion period ends soon and we have made significant efforts to address your concerns with our detailed reply and additional experiments. In particular, we hope that our reply will clarify your major concerns about the scaling effects of the proposed method.
>
> At your convenience, we would greatly appreciate it if you could engage with us and our rebuttal during the remaining time of the discussion so that we can try to address any additional concerns and further improve our manuscript.
>
> Many thanks in advance
>
> Best regards,
>
> The Authors

---

> > ### Comment · Reviewer_mD8g · 2024-11-29
> >
> > Thank you for the author providing a detailed rebuttal and providing experimental results. My question and concern are truly addressed and I'm supporting this paper for acceptance.

---

> > > ### Author Response · Authors · 2024-11-29
> > > **Author response to Reviewer mD8g**
> > >
> > > Thanks for your reply and we are glad that your questions and concerns are truly addressed, and that you're supporting our paper for acceptance!
> > >
> > > We would just like to point out that the rating system seems to be a little different than usual, and it looks like 8 is the score for normal 'acceptance'. If you feel like our paper should be accepted, at your convenience, we would appreciate it if you would consider raising your score to an 8 given the unusual score meanings.
> > >
> > > Many thanks again and wish you a nice day!

---

### Author Response · Authors · 2024-11-17
**Author Response to All Reviewers**

We thank all the reviewers for their insightful comments and helpful suggestions. We hope our responses and paper updates alleviate the concerns raised.

We are happy that our work is conceived as a “novel” (Se9h/mD8g), “flexible” (mD8g), and “exciting” (dQvo) tuning-free alignment approach to “various” tasks (mD8g) and “multiple datasets” (Se9h). We appreciate the reviewers perceive our experimental study as “comprehensive/robust” (Se9h) with “competitive/superior” (mD8g) and “strong” (Se9h) performance.

Following the reviewers’ feedback, we have updated our manuscript mainly with the following contexts:

- **More related works** (Section 2.2-Line149): Following the suggestion from dQvo, we discussed more related works that focus on decoding-phase alignment in the updated manuscript.
- **Detailed decoding pipeline** (Section 3.3-Line254): Following the suggestion from Se9h, we added a complete description of the decoding process to improve clarity.
- **Principle curation analysis** (Section 4.1-Line305): Following the feedback from Se9h and dQvo, we demonstrate how to design the task-specific principles in detail.
- **Method generalization** (Section 4.4-Line 426): Following the suggestion from mD8g, we test the proposed method with smaller and larger, more capable models. The results indicate that  OPAD can generalize well and is very effective on a wide range of mid-scale models (e.g., Vicuna-7B and Vicuna-13B), where it effectively “reinforces” principles to improve alignment.
- **Experiments with Llama 3 family** (Section 4.4-Line 438): Following the suggestion from dQvo, we have directly incorporated the experimental results with the Llama 3 family into the manuscript to enhance the robustness of the method and contribution to the field.

We have highlighted the corresponding modifications in the manuscript with color blue for your convenience. Again, thank you for your hard work. We believe your input has already helped improve the paper and look forward to further engaging with you during the discussion. Please see our replies to each of you below.

---

### Meta-Review · Area_Chair_4neJ · 2024-12-19

**Metareview:**

This paper introduces a novel tuning-free approach for aligning language model outputs with human preferences during inference time. The key scientific contribution is a principle-guided reward function derived from a surrogate optimization solution that modifies model predictions to ensure adherence to target principles. Through comprehensive experiments, the authors demonstrate that their method achieves competitive or superior performance compared to existing baselines across both general and personalized alignment tasks. The main strengths are: 1) elimination of expensive fine-tuning/data collection requirements, 2) flexibility across various tasks, and 3) strong empirical results against RLHF/DPO baselines. Key weaknesses include: computational overhead during inference, lack of theoretical justification for superiority over in-context learning, and initial unclear principle design methodology. I would recommend acceptance primarily due to its novel technical approach and strong empirical validation, while addressing an important practical challenge in LLM alignment.

**Additional Comments On Reviewer Discussion:**

During the discussion period, reviewers Se9h and dQvo raised concerns about principle design methodology, computational efficiency comparisons, and experiments with newer models like Llama-3. The authors thoroughly addressed these by: 1) clarifying the principle curation process and adding detailed examples, 2) providing comprehensive efficiency analysis showing OPAD's advantages over DPO when inference set size is less than 4x training set size, and 3) conducting additional experiments with Llama-3 family models that demonstrated OPAD's effectiveness. Se9h adjusted their score positively after receiving clarification about principle design and computational analysis. Reviewer mD8g's concerns about high tie rates and model scaling were addressed through detailed statistical analysis and additional experiments across model sizes, leading to their support for acceptance. The authors' responses and additional experimental validation, particularly with state-of-the-art models, strengthen the paper's contributions and support the recommendation for acceptance.

---

### Decision · Program_Chairs · 2025-01-22

Accept (Poster)